# Observational study of the implementation of telephone advice nursing in Sweden: did callers follow recommendations and did the rate of healthcare visits change?

Amanda Sundberg,[1] Anna Carin Wahlberg,[2] Niklas Zethraeus,[1]
Korinna Karampampa  [3]

[1]Department of Learning, Informatics, Management and Ethics, Karolinska Institutet, Stockholm, Sweden
[2]Department of Neurobiology, Care sciences and Society, Karolinska Institutet, Stockholm, Sweden
[3]Department of Clinical Neuroscience, Karolinska Institutet, Stockholm, Sweden

**Correspondence to**
Dr Korinna Karampampa;
korinna.karampampa@ki.se

## ABSTRACT

**Objectives** Telephone advice nursing is introduced in many countries with one key aim being a reduction of avoidable healthcare visits. The aim of this study was to explore whether callers to a telephone advice nursing service followed the telenurses' recommendations, and whether there was a change in the level and trend of the rate of healthcare visits after the introduction of telephone advice nursing.

**Design** Observational study.

**Setting** Primary and secondary care in Jönköping Region, Sweden.

**Participants** Telephone advice nursing calls, 6:00–23:00, 2014–2015 (n=185 994) and outpatient healthcare visits 2012–2015 (n=6 877 266).

**Primary outcome** Proportion of callers who visited healthcare within the time period advised by the telenurse.

**Secondary outcome** Change in level or trend of the overall rate of healthcare visits per 1000 persons and 4-week period after the introduction of telephone advice nursing, with subgroup analysis for primary and secondary care.

**Results** 77% of callers who were recommended either to visit healthcare within 24 hours or to 'wait and see' followed the recommendations. There was no significant change in level (−5.15; 95% CI −15.80 to 5.50; p=0.349) or trend (−0.24; 95% CI −0.86 to 0.38; p=0.448) of the overall rate of visits per 1000 persons and 4-week period after the introduction of telephone advice nursing. For the rate of primary care visits, an increase in level (8.01; 95% CI 6.36 to 9.66; p<0.001) and trend (1.28; 95% CI 1.17 to 1.39; p<0.001) were observed. For the rate of secondary care visits, a decrease in level (−8.77, 95% CI −14.41 to −3.13; p=0.004) and trend (−1.03, 95% CI −1.35 to −0.71; p<0.001) were observed.

**Conclusions** The introduction of telephone advice nursing may have contributed to a shift in the rate of healthcare visits from secondary to primary care.

## BACKGROUND

A striking development around the Western world is the increased availability of telephone triage and advice services (henceforth

## Strengths and limitations of this study

► Observational study based on patient-level, registry data covering 4 years and over 7 million observations.
► Statistical analysis using interrupted time series analysis with segmented regression, a recommended method when randomisation is not possible.
► No control series was used to control for potential confounders in the interrupted time series analysis.
► Results might not be generalisable to the rest of Sweden or to other countries.

telephone advice nursing), where telephone nurses (henceforth telenurses) provide callers with triage, advice, support, education, referral to adequate care levels, healthcare information and coordinate healthcare resources.[1] Telephone advice nursing has been implemented in the UK, Denmark, Sweden, Australia, the USA, the Netherlands and Canada.[2–4] In Sweden, the telephone advice nursing service 1177 Vårdguiden (henceforth 1177) was implemented successively across the regions between 2003 and 2013.[5] The service has the five minimum components required to ensure patient safety: qualified staff, medically approved guidelines, electronic medical records, staff training programmes and standards.[6 7]

One key aim from policy-makers when implementing telephone advice nursing services is helping care seekers find the appropriate level of care and reducing unnecessary healthcare visits.[4 8–10] Swedish healthcare providers hypothesised that 1177 would contribute to a more efficient healthcare system.[11–13] It was predicted that if callers followed the telenurses' recommendations, the service would impact on the rates of

healthcare visits as a greater proportion of patients would be able to self-care or seek care at more appropriate healthcare levels than they would without access to telephone advice nursing.[11] The anticipated effect from this transition was an overall reduction of costly emergency visits to secondary care, which would free resources in the healthcare system. The rate of primary care visits was expected to remain more or less unchanged but with a more appropriate case mix.[11]

The available evidence worldwide regarding the extent to which callers follow recommendations from telenurses and telephone advice nursing's impact on healthcare visits is inconsistent.[2 4 8 10] Context-specific factors such as healthcare organisation and workplace environment affect the external validity of findings.[8 14] Identifying key policy lessons for any given telephone advice nursing service therefore requires it to be studied specifically.

For 1177, evidence regarding the extent to which callers follow recommendations from telenurses is diverse. The service's own surveys to callers find that 90% follow the recommendations,[15] while a government agency used observational data to find that 55%–78% follow recommendations depending on the advice provided.[16]

Similarly, evidence regarding 1177's impact on the rate of healthcare visits is scarce. Four studies have found telephone advice nursing services in Sweden to redistribute visits to less costly levels of care.[17–20] Two of the studies concerned local predecessors to 1177.[17 18] All studies based their conclusions on self-reported survey data rather than observational data, which possibly limits the strength of the evidence.

To explore the hypothesis that 1177 may have freed resources in the healthcare system through reducing avoidable healthcare visits, the aim of this study was to use observational data to evaluate whether 1177 callers follow the telenurses' recommendations, and whether there was any change in the level and trend of healthcare visits after the introduction of telephone advice nursing.

## METHODS
### Study design and setting
This was an observational study which used patient-level data from the patient administrative system in Jönköping Region, Sweden. The region manages a tax-financed healthcare system with universal coverage for its 345 000 inhabitants.[21–24] In the region, 1177 was made available to all inhabitants on 2 September 2013. Before that, telephone advice nursing was provided by individual hospitals and healthcare centres in the region. When 1177 was introduced, seven regional telephone advice lines were closed, including those to emergency departments and out-of-hours centres. No other large changes to healthcare in the region were carried out in 2012–2015.[21–24]

In contrast to previous advice lines, 1177 was always open and staffed with nurses specifically trained in telephone advice nursing.[7] The service followed national standards in telephone advice nursing, with a key component being

consistent use of the national symptom-based computer decision support system (CDSS). The CDSS contained symptoms with related urgency levels, as well as a referral catalogue of available healthcare services with opening times, addresses as well as any specific rules about which patient groups different healthcare units receive. The nurses identified the callers' main symptom and thereby obtained the urgency level from the CDSS. They thereafter used the catalogue in combination with their knowledge of the healthcare organisation to find care that would meet the callers' needs within the urgency level, given factors such as opening times, geography and accessibility. The region intended 1177 to be a first point of contact with healthcare for inhabitants, along with the options of reading information on the 1177 website, calling a healthcare centre or visiting an emergency department or out-of-hours care centre.[25]

To determine the percentage of callers who follow the recommendations of the telenurse, calls to 1177 were linked to information on any subsequent outpatient healthcare visits (emergency visits, visits that required referrals and visits that did not require referrals). The linkage and subsequent anonymisation were performed by the system administrator using the unique personal identity number that Swedish residents have. Only anonymised patient-level data were used in the study.

To examine whether there was any change in the level and trend of healthcare visits after 1177 was introduced, anonymised patient-level data on all outpatient healthcare visits in the region (emergency visits, visits that required referrals and visits that did not require referrals) were used.

### Definitions and statistical analysis
The first study outcome of whether callers followed recommendations of the telenurse was defined as the proportion of callers who visited healthcare within the time period advised by the telenurse.

To evaluate this outcome, calls to 1177 between 6:00 and 23:00 and any subsequent healthcare visit at any time of day, that took place within 72 hours of the call between 1 January 2014 and 31 December 2015 (n=185 994), were used. Calls that referred patients to healthcare outside of the region were excluded (n=1839). Night-time calls between 23:00 and 6:00 were unavailable for study as they were answered by a neighbouring county council on behalf of Jönköping Region.

Variables included date and time of call, callers' age and sex, the 'urgency level', as well as the name of any healthcare unit visited within 72 hours of the call, and if the visit occurred within 24 hours.

The variable 'urgency level' described how urgent the telenurse found it for the caller to visit healthcare. The variable had five different levels: 'immediately', 'quickly' (meaning 'within a few hours'), 'within 24 hours', 'a weekday in near time' (meaning within a week or a few weeks) and 'wait and see' (meaning no need to visit healthcare). These five levels were collapsed into three

levels since the data did not allow following the caller's subsequent behaviour on a more granular level:

► Visit within 24 hours (including 'immediately', 'quickly' and 'within 24 hours').
► Visit within a few weeks (including 'a weekday in near time').
► 'Wait and see' (including 'wait and see').

Each call was mapped to the data on healthcare visits, providing the proportion of callers that were recommended to visit within 24 hours and did so, as well as the proportion of callers recommended to 'wait and see' that did not visit healthcare within 72 hours of the call. The proportion of callers who followed the recommendation to visit healthcare within a few weeks could not be determined as the data on healthcare visits only stretched to 72 hours after the call.

To evaluate the second study outcome, that is, to determine whether the rate of healthcare visits changed after the introduction of 1177, all outpatient healthcare visits between 23 January 2012 and 20 December 2015 were used (n=6 877 266). Visits made by patients not registered as residents in the region were excluded (n=215 471). Interrupted time series with segmented regression was used to study this outcome because it is a robust statistical method used where randomised controlled trials cannot be performed.[26–29] The central assumption of the method was that the trend of healthcare visits before the introduction of telephone advice nursing would have continued unchanged if the service had not been introduced.

The data covered 51 four-week periods; 21 before 1177 was implemented and 30 after. Variables included date and time of the visit. The series' data points were calculated as visits per 1000 population and 4-week period. The rates were controlled for population growth by using the population size for each given year as the denominator.

Three time series were made, one for all outpatient visits (primary and secondary care) and one for each subgroup of outpatient primary and secondary care visits. To create the subgroups for outpatient primary and secondary care, the healthcare units were categorised according to a definition from the Swedish Agency for Health and Care Services[16]:

► Primary care: primary care centres, physicians' centres, out-of-hours primary care, children's healthcare centres and maternal healthcare centres.
► Secondary care: hospital-based specialist care including emergency care.

Regression models with two segments were fitted to the data, one for the period pre-1177 and one for the period after. Separate levels (intercepts) and trends (slopes) were estimated for each segment.[27] Statistical tests were carried out to estimate changes in the level or trend after the introduction of 1177. To estimate the size of the effect, the models were used to calculate the rates 1 year (12 four-week periods) after 1177 was introduced and the relative changes compared with the rates that would have been if the trend before 1177 was introduced had continued unchanged.

To control for seasonality, harmonic sine and cosine pairs ('Fourier terms') were added to the model.[30] To control for autocorrelation, a moving averages term of order $q$=13 was added using generalised least squares regression with a maximum-likelihood method. Three plausible alternative autocorrelation terms were tested using likelihood-ratio tests. Since the Akaike information criterion did not indicate improvements to the fit of the model, $q$=13 was retained. Nevertheless, some autocorrelation remained.

### Patient and public involvement

Patients and public were not involved in the design and conduct of this study.

## RESULTS

Calls to the telephone advice nursing service represented 5% of outpatient contacts. The proportion that followed the recommendation differed between groups. A majority of those recommended to visit healthcare within 24 hours did so (73%), and 84% of those recommended to 'wait and see' did not visit healthcare within 72 hours. Combining these groups, 77% followed the recommendation. When stratified by age groups, sex and recommendation, the proportion that followed the tele-nurses' recommendation varied between 66% (females aged 20–39 recommended to visit healthcare within 24 hours) and 90% (children aged 0–9 recommended to 'wait and see') (table 1).

Overall, for all healthcare visits, the final segmented regression model showed no significant changes in level (−5.15; 95% CI −15.80 to 5.50; p=0.349) or trend (−0.24; 95% CI −0.86 to 0.38; p=0.448) of the healthcare visits after the introduction of 1177. Statistically significant changes after the introduction of 1177 were found only in the subgroup analysis. Immediately after the introduction of 1177, primary care visits had an increase of 8.01 visits per 1000 persons (95% CI 6.36 to 9.66; p<0.001) and had a trend increase of 1.28 visits per 1000 per 4-week period (95% CI 1.17 to 1.39; p<0.001). In contrast, visits to secondary care decreased with 8.77 visits per 1000 weeks immediately after the introduction of telephone advice nursing (95% CI −14.41 to −3.13; p=0.004) and had a trend decrease of 1.03 visits per 1000 per 4-week period (95% CI −1.35 to −0.71; p<0.001) (table 2 and figure 1).

When the regression models were used to estimate the effect size 1 year (12 four-week periods) after the introduction of 1177, an 11.55% increase in the rate of primary care visits and an 11.11% decrease in the rate of secondary care visits were found (table 3).

## DISCUSSION

To summarise, the findings of this study indicate that large proportions (77%) of those who call 1177 follow the recommendations, and that there was a shift in healthcare visits from secondary to primary care after the

**Table 1** Callers that received recommendations to visit healthcare or 'wait and see', Jönköping Region 2014–2015

| | | Followed the recommendation to visit healthcare within 24 h n (%) | | Followed the recommendation to 'wait and see' n (%) | | Both groups combined n (%) | |
|---|---|---|---|---|---|---|---|
| | All (n=162 614) | 75 159 | (73) | 50 836 | (84) | 125 995 | (77) |
| | Males (n=72 180) | 34 403 | (75) | 22 533 | (86) | 56 936 | (79) |
| | Females (n=90 434) | 40 756 | (72) | 28 303 | (83) | 69 059 | (76) |
| All | Age group | Followed the recommendation to visit healthcare within 24 h n (%) | | Followed the recommend-action to 'wait and see' n (%) | | Both groups combined n (%) | |
| | 0–9 (n=51 412) | 18 925 | (70) | 22 045 | (90) | 40 970 | (80) |
| | 10–19 (n=16 032) | 8559 | (79) | 4388 | (84) | 12 947 | (81) |
| | 20–39 (n=43 489) | 18 605 | (68) | 13 273 | (82) | 31 878 | (73) |
| | 40–59 (n=25 289) | 13 972 | (77) | 5624 | (80) | 19 596 | (77) |
| | >60 (n=26 392) | 15 098 | (79) | 5505 | (77) | 20 603 | (78) |
| Males | Age group | | | | | | |
| | 0–9 (n=27 562) | 10 246 | (71) | 11 769 | (90) | 22 015 | (80) |
| | 10–19 (n=7365) | 4065 | (81) | 2857 | (84) | 6044 | (82) |
| | 20–39 (n=15 488) | 7232 | (71) | 4496 | (84) | 11 728 | (76) |
| | 40–59 (n=10 706) | 6223 | (78) | 2207 | (80) | 8430 | (79) |
| | >60 (n=11 059) | 6637 | (80) | 2082 | (76) | 8719 | (79) |
| Females | Age group | | | | | | |
| | 0–9 (n=23 850) | 8679 | (70) | 10 276 | (90) | 18 955 | (79) |
| | 10–19 (n=8667) | 4494 | (78) | 2409 | (83) | 6903 | (80) |
| | 20–39 (n=28 001) | 11 373 | (66) | 8777 | (81) | 20 150 | (72) |
| | 40–59 (n=14 583) | 7749 | (76) | 3417 | (79) | 11 166 | (77) |
| | >60 (n=15 333) | 8461 | (78) | 3423 | (77) | 11 884 | (78) |

introduction of 1177. It is possible that the introduction of 1177 contributed to these changes, enabling more patients to seek care at lower levels than they could prior to the service's introduction.

The observed changes in the rate of healthcare visits broadly support the findings of previous survey-based studies in Sweden, which have suggested that telephone advice nursing services in the Swedish setting reduce emergency visits.[17–19] The interpretation that 1177 contributed to the observed changes in the rates of healthcare visits is also strengthened by the lack of plausible alternative explanations: the supply of primary and secondary care

**Table 2** Results of the interrupted time series analysis for healthcare visits, Jönköping Region 2012–2015

| | Level (intercept) | Trend (slope) | Change in level after telephone advice nursing | Change in trend after telephone advice nursing |
|---|---|---|---|---|
| All visits (primary and secondary care) | | | | |
| Coefficient | 376.909 | 0.769 | −5.152 | −0.244 |
| 95% CI | 368.501 to 385.317 | 0.058 to 1.480 | −15.797 to 5.497 | −0.863 to 0.383 |
| P value | <0.001 | 0.040 | 0.349 | 0.448 |
| Primary care visits | | | | |
| Coefficient | 240.649 | −1.164 | 8.013 | 1.279 |
| 95% CI | 238.738 to 242.242 | −1.297 to −1.031 | 6.362 to 9.663 | 1.169 to 1.390 |
| P value | <0.001 | <0.001 | <0.001 | <0.001 |
| Secondary care visits | | | | |
| Coefficient | 142.861 | 1.437 | −8.767 | −1.031 |
| 95% CI | 138.594 to 147.128 | 1.074 to 1.800 | −14.413 to −3.127 | −1.349 to −0.711 |
| P value | <0.001 | <0.001 | 0.004 | <0.001 |

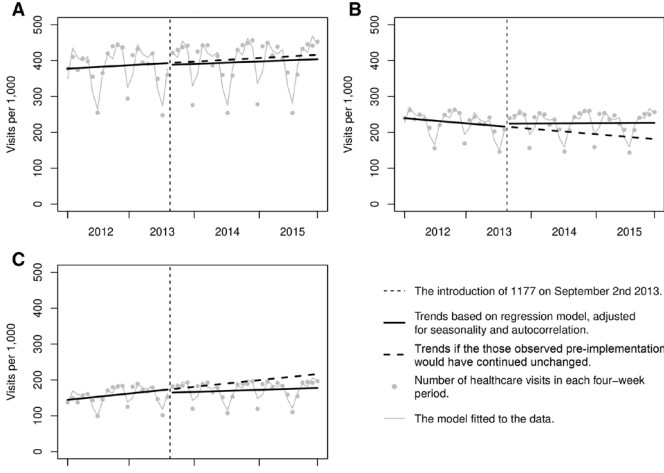

**Figure 1** Interrupted time series analysis for the rate of healthcare visits, Jönköping Region 2012–2015. (A) All visits (primary and secondary care). (B) Primary care visits. (C) Secondary care visits. In these graphical displays of the interrupted time series analyses, the trends of the rates of healthcare visits pre-implementation and post-implementation of 1177 are shown (the thick solid lines). The pre-implementation trend has been extrapolated to the post-implementation period (the thick dotted line), to show what would have happened if the pre-implementation trends would have continued unchanged. The data points used for the analysis are shown in grey. The model fitted to the data show the seasonal trends (the thin grey lines).

was stable throughout the study period and no other potential confounders were found.[21–24]

Compared with 1177's own survey-based findings on the proportions that follow the recommendations of the telenurse (90%), 77% is lower.[15] This is also the case when comparing with earlier survey-based studies on local predecessors to 1177.[1 17 18 31] However, the finding is similar to those of a government agency report that used observational data.[16] This contrast between survey data and observational data is supported by systematic reviews of the international literature on telephone advice nursing, according to which self-reported data tends to provide higher proportions than observational data.[2]

These inconsistent results of survey and observational data respectively may be considered as complementary as they to some extent measure different constructs. While

observational studies require researchers to define what it means to follow the recommendations of the telenurse, survey-based studies let respondents interpret its meaning subjectively.[32] Although the latter study design suffers from threats to validity, primarily from recall bias, social desirability bias and non-response bias,[33–35] it also allows for capturing nuances of what it means to follow recommendations, and for studying the patient's perspective.

There are a broad range of possible care-seeking scenarios concealed behind the proportions found in this study. First, when callers appear to have followed the recommendations, it is not certain that the call to 1177 was what led them to do what they did. They may have based their decisions on consultations with primary care, family and friends, websites and so forth. Second, the data in this study only showed the telenurse's main recommendation and cannot reflect potential alternative courses of action that the telenurse and caller may have also discussed, for example, depending on how the patient's health status developed. Third, care sought after a call to 1177 within the time recommended by the nurse may have been for a separate issue than that discussed with the telenurse. Fourth, care recommended by the telenurse may have been inaccessible to the caller, making it impossible to follow the recommendations. Finally, callers may have chosen not to follow the recommendations because they lacked confidence in the telenurse or found the recommendations inappropriate.[20 36 37]

### Strengths and limitations of the study

The strength of the study is that complete anonymised patient-level data were used, providing a strong design for answering the research questions. Unlike previous studies, this study's findings were derived from a large number of observations, free from the biases of surveys.[33] The interrupted time series analysis was the next-best statistical method for studying the rate of healthcare visits before and after the introduction of 1177 given a randomised, controlled trial was not possible.[27–29]

This study also has several limitations. Ideally, a control series should have been used to control for potential confounders.[27] In lack of a control group, media archives and grey literature were searched to identify potential confounding events, but none were found.[21–24]

**Table 3** Observed rates of healthcare visits 12 four-week periods after telephone advice nursing was introduced and rates if the trends observed pre-implementation would have continued unchanged

| | All visits (primary and secondary care) (p=0.448) | Primary care visits (p<0.001) | Secondary care visits (p<0.001) |
|---|---|---|---|
| Observed rate per 1000 persons | 394.20 | 225.59 | 169.13 |
| Rate per 1000 persons if the trend observed pre-implementation would have continued unchanged | 402.28 | 202.23 | 190.27 |
| Absolute change (observed rate–rate if the trend observed pre-implementation would have continued unchanged) | −8.02 | 23.36 | −21.14 |
| Relative change (%) | −2.01 | 11.55 | −11.11 |

The autocorrelation that remained in the segmented regression model fitted to the data, despite attempts to control for it, is another limitation. Autocorrelation does not affect coefficients, but it does affect the SEs, which means we cannot rely on the p values. This is particularly the case when p values are close to 0.05. In the results of this study, however, p values for most statistically significant results were <0.001.

Finally, a limitation concerns the generalisability of the results. First, this study has focused on 1177 in one Swedish region. While the healthcare organisation across the regions in Sweden do not differ substantially, some differences may be present regarding the availability of healthcare services and the population demographics. Therefore, the results might not be generalisable to the rest of Sweden, or to other countries.

In addition, due to the unavailability of data, night calls during the hours 23:00 and 6:00 were excluded from this study. However, it is unlikely that the behaviour of night-time callers would be substantially different than that of people calling during the rest of the day. Still, since the level of urgency may be different between, for example, morning or night calls, the results of this study may be not generalisable to calls/callers throughout the entire day.

### Implications for future research and policy

For policy-makers, this study provides some support that the implementation of telephone advice nursing contributed to their aims of freeing resources in the healthcare system, as a large proportion of callers followed the nurses' recommendations, and as the service may have contributed to the change in the rate of healthcare visits. The implication for policy is that telephone advice nursing might be considered as a positive element in the healthcare organisation.

Three areas should be of specific interest to future research and policy. One is the effects of technology used in telephone advice nursing: given the rapid technological developments such as automatic triage and the use of photos and video calls in telephone advice nursing, it should be of interest to find out more about how technology is being used and can be used to develop and enhance telephone advice nursing and its role in the healthcare system.[8]

Another area to further study is how the interaction between telephone advice nursing and other services in the healthcare system affects efficiency. For example, studies have established that one of the largest challenges experienced by telenurses at 1177 in Sweden is the lack of adequate resources in the healthcare organisation, with insufficient availability of primary care being the largest problem.[3 6 38–43] A recent study has also shown that the healthcare organisation is not adapted to meet the needs of callers with mental illness.[44] If the healthcare organisation is not set up to meet the needs of the population, 1177 cannot contribute to system efficiency with its full potential. Future studies of 1177's contribution to system efficiency should treat it not as an isolated service but as one whose efficacy is interlinked with the functioning of the healthcare system as a whole.[45]

In addition, future research may focus on how access to, and use of, telephone advice nursing varies in the population. This study found that only 5% of outpatient healthcare contacts were calls to 1177, raising the question of whether more patients could benefit from using the service. This aspect has not been studied in Sweden, but studies in the UK have found that the former telephone advice nursing service National Health Service Direct was used more by some groups. Important determinants were ethnicity, gender, age and socioeconomic status.[46–49] If telephone advice nursing can contribute to inhabitants receiving care at the most appropriate levels, it is important that all groups gain equal access to the service.

## CONCLUSIONS

The results indicate that large proportions of those who call 1177 follow the recommendations and that the introduction of 1177 in Region Jönköping may have contributed to a shift in healthcare visits from secondary to primary care. The results provide some support to the hypothesis that 1177 may have contributed to system efficiency, freeing resources through reducing unnecessary costly healthcare visits.

**Contributors** AS, KK and NZ contributed to the conceptualisation of the research questions, study design and methods. AS performed the analysis of the data, interpreted the study findings and drafted the manuscript. All remaining authors assisted in the interpretation of the study findings and contributed to the manuscript with comments, suggestions and text.

**Funding** The authors have not declared a specific grant for this research from any funding agency in the public, commercial or not-for-profit sectors.

**Competing interests** AS is employed by Inera AB, a limited company owned by Swedish regions, municipalities as well as the Swedish Association of Local Authorities and Regions (SALAR) that administrates the technical solutions used in 1177. AS conducted this research as part of her master's degree project in the Master's programme in Public Health Sciences (track Health Economics, Policy and Management) at the Karolinska Institutet, while she was on leave of absence from her employment. The study findings have been interpreted scientifically by all coauthors, without the involvement of Inera AB, and not in the interest of 1177. All other coauthors (ACW, NZ, and KK) are employed at or affiliated with Karolinska Institutet, Stockholm, Sweden. KK is currently employed by BMS, a pharmaceutical company which is not involved with 1177 in Sweden.

**Patient and public involvement** Patients and/or the public were not involved in the design, or conduct, or reporting, or dissemination plans of this research.

**Patient consent for publication** Not required.

**Ethics approval** Informed consent was not collected from the participants of this study. The study was approved by the Swedish Ethical Review Authority, registered under number 2019-01784.

**Provenance and peer review** Not commissioned; externally peer reviewed.

**Data availability statement** Data may be obtained from a third party and are not publicly available. The data that support the findings of this study are available from Region Jönköping, but restrictions apply to the availability of these data, which were used under license for the current study, and so are not publicly available.

ORCID iD
Korinna Karampampa http://orcid.org/0000-0002-2578-1865

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
