## [Reviewer comments · BMJ Open]

ARTICLE DETAILS

TITLE (PROVISIONAL)	An observational study of the implementation of telephone advice nursing in Sweden: did callers follow recommendations and did the rate of healthcare visits change?
AUTHORS	Sundberg, Amanda; Wahlberg, Anna Carin; Zethraeus, N; Karampampa, Korinna

VERSION 1 – REVIEW

REVIEWER	Hardie, Rae-Anne Macquarie University Faculty of Medicine and Health Sciences, Centre for Health Systems and Safety Research, Australian Institute of Health Innovation
REVIEW RETURNED	29-Mar-2021

GENERAL COMMENTS	Thank you for the opportunity to review this nicely written paper, it was a great read. Your study was interesting in that it contrasted what self-reported survey data said about rates of following recommendations, which is often the case with self-reported data- this clarification is important for policymakers and for quality improvements to the system. I have a few comments and questions that would clarify a few uncertainties: Methods: -It would be helpful to know the availability of telenursing hours to better understand why the hours of 6 am-11 pm were chosen for the purposes of this study, as the exclusion of night calls could bias results.-Did urgency levels make any difference in advice for type or level of care that nurses recommended should be sought (primary vs secondary)? What guidelines were nurses following that informed their recommendations/their advice?-It would be nice to see an explanation for why interrupted time series was selected in the study design.-Was tele nursing implemented at the same time point for all groups?-How were rates controlled for population growth? What about differences in demographic factors? Results: -Period missing in line 211 after $p < 0.001$-Regression: was increase in primary and decrease in secondary care consistent over the year, or did this differ by period post-implementation?-If calls to telenursing only accounts for 5% of outpatient care, aim 2 is then going to be difficult to quantify based on rates overall- how can we know whether decreases or improvements are directly due to telenurse interventions/advice? Were there other
---

	widespread public education campaigns around the time of the implementation or any other changes made during the rollout that could also have impacted care type? -What is meant by counterfactual in Table 3? I can't find any other mention of it in the paper, if this wording is chosen then it should be explained elsewhere. Discussion -Incorrect whose (who's) in line 305. -Another possible limitation- could care seeking post-telenurse call within the study period potentially sometimes be about a separate issue not discussed with the telenurse? -I am curious about advice that falls just before a weekend or holiday periods- have you noticed any trends when access to certain healthcare services (e.g. getting in to see a primary care doctor when their practice may be closed within 24 hours might be difficult) might result in lower "advice-following"?
--	---

REVIEWER	Liang, Jingyan Yangzhou University
REVIEW RETURNED	07-May-2021

GENERAL COMMENTS	General Comments and Overall Evaluation Information about implementation of telenursing is very important for researchers who are interested in telenursing intervention for persons' health. This manuscript attempted to explore whether callers to a telenursing service followed the telenurses' recommendations, and whether there was a change in the level and trend of the rate of healthcare visits after the introduction of telenursing. The author(s) concluded that the introduction of telenursing may have contributed to a shift in the rate of healthcare visits from secondary to primary care. However, there are some issues that need to be addressed. 1.First of all, this manuscript used the data of 2012-2015. In the information age, the attitude and frequency of telenursing use of patients is changing greatly every year, even every day. Especially after covid-19, the usage of telenursing in most countries is obviously different from that in the past, using 2015 data for analysis is a bit of a lag. 2.The title of the article is "Implementation of Telenursing". Telenursing covers many aspects, but the article only introduces telephone advice nursing. I think the title of the article is inconsistent with the content. The content of the article should be much richer. 3.Because the paper was just investigated in Jönköping Region in Sweden, as the author said, "Results might not be generalizable to the rest of Sweden, or to other countries". Perhaps the authors should broaden their investigation. 4.In the Methods section, authors did not introduce the statistical methods. Please clarify it. 5.The article should describe and discuss the implications of the study, rather than merely explain the data collected, as the reader may not understand how the results will be of value to future research or nursing work. 6.The references need to be updated.
---

REVIEWER	You, Zhiying University of Colorado Denver School of Medicine, Medicine
REVIEW RETURNED	17-May-2021

GENERAL COMMENTS	Very interesting study and well written manuscript. Page 4, lines 38-40, unclear why primary and secondary outcomes were mentioned in a section of participants. Page 9, line 140-142, please include the number of these calls if available. Page 27, figure 1, please add footnote to guide the readers in reading the figure.
--

REVIEWER	McDaniel, Lee LSU Health Sciences Center New Orleans, School of Public Health
REVIEW RETURNED	28-May-2021

GENERAL COMMENTS	The authors have presented a manuscript that is, in my experience, uncharacteristically sound from a statistical perspective given the topic. The basic technique seems fundamentally sound and the authors are appropriately restrained in their conclusions. It would be nice to have a control group for this study, but it would also be nice to have a big pile of cash to dive into. I would be interested to know if there were any other attempts to improve the healthcare system in Sweden at this time. Obviously this study will always have bring up these kinds of questions. In the limitations part of the discussion section, I think the authors may be able to do better than "it is not known to what extent [the autocorrelation] affected the results". I think the authors may be able to say something about what happens. Will the estimates still be consistent? As a final note, I would point out that "Akaike" is a person's name and should be capitalized.
---

REVIEWER	Matsouaka, Roland Duke Clinical Research Institute, Biostatistics and Bioinformatics
REVIEW RETURNED	04-Jun-2021

GENERAL COMMENTS	The implementation of telenursing in Sweden: an observational study focusing on whether callers follow recommendations, and if there is a change in the rate of healthcare visits. Authors: Amanda Sunberg et al. In this paper, the authors conducted an observational study where they investigated whether telenursing calls were followed by outpatient healthcare visits in one region in Sweden after the introduction of the 1177 Vardguiden. To do so they consider a fairly robust inclusion exclusion criteria of the participants in the study and leverage interrupted time series with segmented regression. The paper is well-written and the authors have done a great job elaborating on the different strengths and weaknesses of their study. I don't have major comments, but just minor comments for clarification. Minor points:  1. Remove the extra "there" in the title. The title should read as I wrote here. 2. On page 4, line 38-45: it is not clear what are the participants, the primary outcomes, or the secondary outcomes. I
--

	can't just choose from this list of outcomes provided here. Do the other consider participants as outcomes? This is clearly stated in the manuscript, but somehow it is unclear here.
--	---

VERSION 1 – AUTHOR RESPONSE

Responses to comments from Reviewer 1

Dr. Rae-Anne Hardie, Macquarie University Faculty of Medicine and Health Sciences

Thank you for the opportunity to review this nicely written paper, it was a great read. Your study was interesting in that it contrasted what self-reported survey data said about rates of following recommendations, which is often the case with self-reported data- this clarification is important for policymakers and for quality improvements to the system. I have a few comments and questions that would clarify a few uncertainties:

Methods:

Comment 1: It would be helpful to know the availability of telenursing hours to better understand why the hours of 6 am-11 pm were chosen for the purposes of this study, as the exclusion of night calls could bias results.

Author's response: Telenursing was available to inhabitants at all times throughout the day. We have added this information in the methods section.

Night-time calls between 11 pm and 6 am were answered by a neighboring county council on behalf of Jönköping Region, resulting in their data being registered in the system of the neighboring county council and thereby unavailable for this study. We have added this information to the methods section so that the reader knows why the hours were chosen.

Since we haven't been able to study the night calls, we don't know if the extent to which callers followed the advice differed in night calls compared to the day calls in the study, which would make our results less generalizable to calls happening throughout the day. We have added this clarification in the discussion section of the manuscript.

Comment 2: Did urgency levels make any difference in advice for type or level of care that nurses recommended should be sought (primary vs secondary)? What guidelines were nurses following that informed their recommendations/their advice?

Author's response: Thank you for your questions. Nurses follow a clinical decision support system, which helps them to first narrow down to which symptom the caller is calling for. Every symptom in the clinical decision support system provides the nurse with an urgency level. However, after that, it is up to the nurse to decide which type and level of care the caller should be recommended. The clinical support system provides the nurse with a catalogue of all health care available in Sweden, with opening times, addresses as well as any specific rules about which patient groups the health care units receive. The nurse can use the catalogue to find suitable health care, but the final recommendation depends on what is open within the limits of the urgency level and on the nurse's knowledge and understanding of the caller's symptoms, needs, and of the health care organization.

We have revised the 'setting' part of the methods section to describe what guidelines nurses were following that informed their recommendations.

Comment 3: It would be nice to see an explanation for why interrupted time series was selected in the study design.

Author's response: Interrupted time series was selected because it is the most robust research design where randomized controlled trials are not possible, as was the case in this study where telephone advice nursing was implemented directly in its real-world setting. We have added a sentence about this in the manuscript.

Comment 4: -Was tele nursing implemented at the same time point for all groups?

Author's response: Yes, telenursing was implemented for all inhabitants in Jönköping Region at the same time point. It was a population wide intervention. We have added this clarification in the methods section.

Comment 5: -How were rates controlled for population growth? What about differences in demographic factors?

Author's response: The rates were controlled for population growth by using the population size for each year as the denominator. We have added this information to the methods section.

With regards to other demographic factors (i.e. age and sex), in the interrupted time series analysis, there was no need to control for those (age and sex) as our aim was to study the intervention which was aimed toward the entire population regardless of age and sex. So, neither age nor sex could be confounders in the interrupted time series analysis, as we are looking at telenursing's effects on healthcare use as a whole, regardless of if use differs between groups. Future research could focus on subgroup analysis for age, sex or socioeconomic status, which could provide further knowledge about the role of telephone advice nursing in the healthcare system. This clarification is added in the manuscript.

Results:

Comment 6: -Period missing in line 211 after $p < 0.001$

Author's response: Thank you for alerting us, we have changed this.

Comment 7: Regression: was increase in primary and decrease in secondary care consistent over the year, or did this differ by period post-implementation?

Author's response: Thank you for your comment; interrupted time series analysis used in this study focused on long-term trends, and therefore cannot tell us anything about within-period analysis, i.e. differences within periods of a year

Comment 8: -If calls to telenursing only accounts for 5% of outpatient care, aim 2 is then going to be difficult to quantify based on rates overall- how can we know whether decreases or improvements are directly due to telenurse interventions/advice? Were there other widespread public education campaigns around the time of the implementation or any other changes made during the rollout that could also have impacted care type?

Author's response: Thank you for your comment. Yes, indeed it is difficult to be certain if the changes in the rates of healthcare use are directly due to the implementation of telephone advice nursing from this study. As it is pointed out, it is not possible to rule out that other events than the implementation of 1177 could have affected the rate of healthcare visits during the study period. The use of a control group in this study would have helped minimize this threat to validity.

However, due to availability of data, the use of such a control group was not possible. In the absence of a control group, media archives and grey literature were searched to identify potential confounding events. No other public education campaigns or changes to the healthcare system were found. The supply of healthcare in the region was relatively stable throughout the study period, with an increase in physicians, a decrease in nurses and a small decrease in primary health care centres – small changes that are unlikely to contribute to the change in rates observed here. For example, the decrease in primary health care centres might be expected to mean that primary care becomes less accessible so that more patients are forced to seek secondary care and thereby counteract the changes observed in the study.

We have revised throughout the manuscript to clarify what we know about potential confounders. Since we cannot exclude the possibility that other events than the implementation of 1177 caused the changes in the rates observed, we are restrictive in our conclusions and say only that the introduction of telephone advice nursing may have contributed in a shift in the rate of healthcare visits from secondary to primary care.

Comment 9:-What is meant by counterfactual in Table 3? I can't find any other mention of it in the paper, if this wording is chosen then it should be explained elsewhere.

Author's response: We have replaced 'counterfactual' with "Rate per 1,000 persons if the trend observed pre-implementation would have continued unchanged" in the title and contents of Table 3 to better clarify this. We also found the term counterfactual in the figure 1 and have replaced it there as well.

Discussion

Comment 10: -Incorrect whose (who's) in line 305.

Author's response: Thank you for this comment, we have changed this.

Comment 11: -Another possible limitation- could care seeking post-telenurse call within the study period potentially sometimes be about a separate issue not discussed with the telenurse?

Author's response: Yes, of course – thank you for this suggestion, we have added it to the manuscript.

Comment 12: -I am curious about advice that falls just before a weekend or holiday periods- have you noticed any trends when access to certain healthcare services (e.g. getting in to see a primary care doctor when their practice may be closed within 24 hours might be difficult) might result in lower "advice-following"?

Author's response: Thank you for your comment. This is indeed a very interesting question. Due to data availability, and the fact that this is one of the first studies of its kind, the aim of our study was to first understand the use of the service and recommendations regardless of the day they were made, and not in connection with factors like availability of primary and secondary healthcare during e.g. the holiday season. Since this is a relevant question, future research should focus on disentangling the effect of such availability from the use and follow of recommendations of the 1177 service.

Responses to comments from Reviewer 2

Dr. Jingyan Liang, Yangzhou University

General Comments and Overall Evaluation

Information about implementation of telenursing is very important for researchers who are interested in telenursing intervention for persons' health. This manuscript attempted to explore whether callers to a telenursing service followed the telenurses' recommendations, and whether there was a change in the level and trend of the rate of healthcare visits after the introduction of telenursing. The author(s) concluded that the introduction of telenursing may have contributed to a shift in the rate of healthcare visits from secondary to primary care. However, there are some issues that need to be addressed.

Comment 1: First of all, this manuscript used the data of 2012-2015. In the information age, the attitude and frequency of telenursing use of patients is changing greatly every year, even every day. Especially after covid-19, the usage of telenursing in most countries is obviously different from that in the past, using 2015 data for analysis is a bit of a lag.

Author's response: Thank you for your comment. It is true that telenursing is changing every day. Today, in 2021, telenursing is being developed and improved with new digital resources for both telenurses and patients, for example with automatic triage and anamnesis tools prior to the call and photo and video technology to enhance communication between the nurse and the caller. And of course, these changes are important to consider in research and policy today and in the future. We have added a section about this under the section 'Implications for future research and policy'.

However, we would like to point out that regardless of the availability of the tools we have now for these type of services, what we attempt to study here is related to the use of the service, if callers followed the advice and the impact this had on healthcare use; we do not believe that any of these outcomes would be influenced substantially by the tools used for telenursing (digital/video vs phone communication). Therefore, our results could be considered relevant for nowadays use of telenursing as well. There is still limited knowledge about the role of telephone advice nursing in health care systems and this study can provide such a contribution and constitute one stepping stone for further research. It might also still be of interest to policy makers and researchers who seek to understand and learn about the consequences from changes made to the health care system.

Comment 2: The title of the article is "Implementation of Telenursing". Telenursing covers many aspects, but the article only introduces telephone advice nursing. I think the title of the article is inconsistent with the content. The content of the article should be much richer.

Author's response: 'Telenursing' is indeed a broad term that could include more than nursing advice over the phone. We changed 'telenursing' to 'telephone advice nursing' throughout the manuscript to be clearer about what we mean. We also use the term in the responses to comments in this letter.

Comment 3: Because the paper was just investigated in Jönköping Region in Sweden, as the author said, "Results might not be generalizable to the rest of Sweden, or to other countries". Perhaps the authors should broaden their investigation.

Author's response: While a broader investigation would be very interesting, we unfortunately had no access to data outside the Jönköping Region in Sweden. This limitation is mentioned in the discussion section of our manuscript.

Comment 4: In the Methods section, authors did not introduce the statistical methods. Please clarify it.

Author's response: Thank you for your comment. We have edited the description of the statistical methods we had in our manuscript to make sure our methods are clear to the readers.

For study outcome one, proportions were calculated for callers who followed the recommendations to visit healthcare within 24 hours and to 'wait and see', respectively. For study outcome two, we used interrupted time series with segmented regression.

Comment 5: The article should describe and discuss the implications of the study, rather than merely explain the data collected, as the reader may not understand how the results will be of value to future research or nursing work.

Author's response:

Indeed, this is important. In the introduction of the article, we introduce the key aims that policymakers had when introducing telephone advice nursing, in helping care seekers find the appropriate level of care and reducing avoidable healthcare visits, which were hoped to free resources in the healthcare system.

In the discussion, we discuss the findings and provide some support to the hypothesis that the policy makers had when deciding to implement the service.

This is what we can discuss in terms of implications based on the data we had, our methods, and findings. To further clarify the value of the study we have added a paragraph under the section 'implications for future research and policy', where we write that the study provides some support to policy makers to consider forms of telephone advice nursing as an element in the healthcare organization.

Comment 6: The references need to be updated.

Author's response: At the time of submission to the journal we had the latest references; now, we have updated our literature search to include recent studies, where relevant.

Responses to comments from Reviewer 3

Dr. Zhiying You, University of Colorado Denver School of Medicine, Colorado School of Public Health

Comments to the Author:

Very interesting study and well written manuscript.

Comment 1: Page 4, lines 38-40, unclear why primary and secondary outcomes were mentioned in a section of participants.

Author's response: Thank you, we have removed mention of primary and secondary outcomes in the section of participants.

Comment 2: Page 9, line 140-142, please include the number of these calls if available.

Author's response: Thank you, we have added the number of calls that were excluded to the manuscript (n=1,839). We have also moved mention of excluded visits from this line to the section on the second study outcome, where it belongs, and added the number of excluded visits (n=215,471).

Comment 3: Page 27, figure 1, please add footnote to guide the readers in reading the figure.

Author's response: We have added a footnote that we hope can guide readers in reading the figure.

Responses to comments from Reviewer 4

Dr. Lee McDaniel, LSU Health Sciences Center New Orleans

Comment 1: The authors have presented a manuscript that is, in my experience, uncharacteristically sound from a statistical perspective given the topic. The basic technique seems fundamentally sound and the authors are appropriately restrained in their conclusions. It would be nice to have a control group for this study, but it would also be nice to have a big pile of cash to dive into.

Author's response: Thank you. Yes, indeed, a control group would have been nice to have, unfortunately we did not have that opportunity. Relevant text is added in the discussion section of the manuscript.

Comment 2: I would be interested to know if there were any other attempts to improve the healthcare system in Sweden at this time. Obviously this study will always have bring up these kinds of questions.

Author's response: Indeed, we briefly mention this in the discussion, and we have revised the manuscript throughout to clarify what is known about other attempts to improve the healthcare system at the time.

Comment 3: In the limitations part of the discussion section, I think the authors may be able to do better than "it is not known to what extent [the autocorrelation] affected the results". I think the authors may be able to say something about what happens. Will the estimates still be consistent?

Author's response: Thank you. We have added relevant text in our discussion to describe the consequences of autocorrelation.

Comment 4: As a final note, I would point out that "Akaike" is a person's name and should be capitalized.

Author's response: Thank you for pointing this out, we have changed this.

Responses to comments from Reviewer 5

Dr. Roland Matsouaka, Duke Clinical Research Institute

Authors: Amanda Sunberg et al. In this paper, the authors conducted an observational study where they investigated whether telenursing calls were followed by outpatient healthcare visits in one region in Sweden after the introduction of the 1177 Vardguiden. To do so they consider a fairly robust inclusion exclusion criteria of the participants in the study and leverage interrupted time series with segmented regression. The paper is well-written, and the authors have done a great job elaborating on the different strengths and weaknesses of their study. I don't have major comments, but just minor comments for clarification. Minor points:

Comment 1: Remove the extra “there” in the title. The title should read as I wrote here.

Author’s response: Thank you for noticing this. As the editor asked us to shorten the title, we have deleted the repeated word and edited further the title.

Comment 2: On page 4, line 38-45: it is not clear what are the participants, the primary outcomes, or the secondary outcomes. I can’t just choose from this list of outcomes provided here. Do the other consider participants as outcomes? This is clearly stated in the manuscript, but somehow it is unclear here.

Author’s response: Thank you for mentioning this. We have clarified it and removed the mention of outcomes in this part.

VERSION 2 – REVIEW

REVIEWER	Hardie, Rae-Anne Macquarie University Faculty of Medicine and Health Sciences, Centre for Health Systems and Safety Research, Australian Institute of Health Innovation
REVIEW RETURNED	15-Jul-2021

GENERAL COMMENTS	The authors have done an excellent job in addressing all of the reviewers' comments to the best of their reasonable ability. I recommend this article to be published. Great work!
--

REVIEWER	McDaniel, Lee LSU Health Sciences Center New Orleans, School of Public Health
REVIEW RETURNED	07-Jul-2021

GENERAL COMMENTS	Your revisions have adequately addressed my concerns. However, I now have a small new concern. You've referred to the segmented time-series as a "study design" but this isn't quite right. The segmented time-series is really an analysis strategy. A good way to think of this distinction, I think, is that design elements are things you can't change at the manuscript stage without changing the underlying dataset. Analysis elements can be changed without doing anything to the dataset.
--

VERSION 2 – AUTHOR RESPONSE

Responses to comment4 from Reviewer 4

Dr. Lee McDaniel, LSU Health Sciences Center New Orleans

Comment 1: Your revisions have adequately addressed my concerns. However, I now have a small new concern. You've referred to the segmented time-series as a "study design" but this isn't quite right. The segmented time-series is really an analysis strategy. A good way to think of this distinction, I think, is that design elements are things you can't change at the manuscript stage without changing the underlying dataset. Analysis elements can be changed without doing anything to the dataset.

Author's response: Thank you for raising this question. Of course, we agree with your comment, time-series is a statistical analysis strategy, and not a study design. We have now made minor revisions to the manuscript to further clarify this.